

# SARS-CoV-2 seroprevalence study in Lambayeque, Peru. June–July 2020

Cristian Díaz-Vélez[1,2,*], Virgilio E. Failoc-Rojas[3,*],
Mario J. Valladares-Garrido[4], Juan Colchado[1],
Lourdes Carrera-Acosta[5,6], Mileny Becerra[7], Dafne Moreno Paico[7] and
Elgin Thom Ocampo-Salazar[5,8]

[1] Oficina de Inteligencia Sanitaria, Hospital Nacional Almanzor Aguinaga Asenjo, EsSalud, Chiclayo, Peru
[2] Facultad de Medicina, Universidad Cesar Vallejo, Chiclayo, Peru
[3] Universidad San Ignacio de Loyola, Lima, Peru
[4] Universidad Continental, Lima, Peru
[5] Ministerio de Salud, Lima, Peru
[6] Universidad Nacional Mayor de San Marcos, Lima, Peru
[7] Dirección Regional de Salud Lambayeque, Lambayeque, Peru
[8] Universidad Particular de Chiclayo, Chiclayo, Peru
* These authors contributed equally to this work.

## ABSTRACT

**Background:** Estimating the cumulative prevalence of SARS-COV-2 will help to understand the epidemic, contagion, and immunity to COVID-19 in vulnerable populations. The objective is to determine the extent of infection in the general population and the cumulative incidence by age group.

**Methods:** It was carried out with a longitudinal analytical study, in the population of the Lambayeque region, located in the north of Peru. The selection was carried out in multistages (districts, area, household, and finally choosing the interviewee within the house). Seroprevalence was estimated as a positive result of the rapid test whether it was positive IgM or positive IgG. An adjustment was made for the sampling weights used.

**Results:** The seroprevalence found in the region was 29.5%. Young people between 21 and 50 years old presented the highest seroprevalence frequencies. A total of 25.4% were asymptomatic. The most frequent complaint was dysgeusia and dysosmia (85.3% and 83.6%). Dysosmia (PR = 1.69), chest pain (PR = 1.49), back pain (PR = 1.45), cough (PR = 1.44), fever (PR = 1.41), general malaise (PR = 1.27) were associated factors with the higher the frequency of seropositivity for SARS-CoV-2. Reporting of complete isolation at home decreased the frequency of positivity (PR = 0.80), however, reporting having ARI contact (PR = 1.60), having contact with a confirmed case (PR = 1.51), and going to market (PR = 1.26) increased the frequency of positivity for SARS-CoV-2.

**Conclusion:** These results suggest that Lambayeque is the region with the highest seroprevalence in the world, well above Spain, the United States and similar to a study in India.

Corresponding author
Virgilio E. Failoc-Rojas,
virgiliofr@gmail.com

## INTRODUCTION

On January 30, 2020, the World Health Organization (WHO) declared coronavirus infection 2019 (COVID-19) as a public health emergency of international concern (*World Health Organization, 2020*). To date, nearly 90 million confirmed COVID-19 cases have been reported (Worldometeres). In Peru, the pandemic began on March 6, 2020, since then almost 471 thousand cases have been reported, of which 88.7% are mild and asymptomatic (*Ministry of Health of Peru (MINSA), 2020*).

Previous studies have found that 80.9% of COVID-19 cases have mild disease (*Hu et al., 2020*) and 40–45% are asymptomatic (*Oran & Topol, 2020*) with different clinical courses due to the heterogeneity of the affected population. Asymptomatic cases with COVID-19 have shown active subclinical infection, managing to transmit the virus to other people for a prolonged period (more than 14 days), its identification and characterization being crucial to control outbreaks in various regions (*Hu et al., 2020*; *Oran & Topol, 2020*).

For the detection of SARS-CoV-2, the IgM and IgG antibodies produced from the second week of viral infection are used (*Sethuraman, Jeremiah & Ryo, 2020*), which detect antibodies against the nucleocapsid antigen and have an immunological memory for up to 6 weeks after being transmitted from person to person (*Li et al., 2005*; *Tan et al., 2020*). The use of rapid tests has a false positive rate that is less than 3% (*Lisboa Bastos et al., 2020*).

This is useful in the development of epidemiological studies, where it is required to evaluate the presence of antibodies (IgM or IgG) to know the accumulated prevalence of cases with COVID-19 infection.

Currently, in Peru, the diagnosis of COVID-19 is limited to examining patients who require medical attention for symptoms, or contact of those confirmed patients, which in addition to affecting the presentation of the national and local epidemic curves (*Loyola et al., 2020*) underestimates the real number of patients with asymptomatic or subclinical COVID-19. Estimating the cumulative prevalence of SARS-COV-2 will help to understand the epidemiology of the outbreak, contagion, and immunity to COVID-19 in vulnerable populations (*Munster et al., 2020*), as well as to know the proportion of the population that has already developed antibodies to anticipate its dynamics and plan a public health response appropriately (*Clapham et al., 2020*; *Lipsitch, Swerdlow & Finelli, 2020*).

In this work, a population epidemiological investigation is presented at the Lambayeque region level, through a serological evaluation whose objective is to determine the extent of the infection in the general population and the cumulative incidence by age group, as well as to determine the proportion of asymptomatic infections or subclinical.

## MATERIALS AND METHODS

### Study design and participants

The seroprevalence study was carried out with a prospective, analytical cross-sectional study design of population inference at the level of the Lambayeque region, a region

located in the North of Peru. Lambayeque has a total population of 1,197,260 inhabitants and is divided into three provinces. The population has 48.5% males; of the total population, the population aged 15–64 represents 64% and those over 65 represent 8.8%. The age distribution by sex tends to be similar in both groups (*National Institute of Statistics and Informatics of Peru (INEI), 2018*).

## Sample and sampling

To calculate the sample size, the population size of 1,197,260 inhabitants was taken as a reference (*Informática INdEe, 2018*). The expected proportion of seropositivity in the population of 4.5% (*Sood et al., 2020*) was considered, with a confidence level of 95% and an absolute precision of 0.01 and at the level of three provinces Chiclayo, Ferreñafe, and Lambayeque, stratified by sex, giving a total of 2,010 people. Due to the dynamic condition of the population, it was optimized for its forecast at the date of application of the sampling with the application of simple linear regression analysis and the logarithm growth method.

The selection was carried out in multistages, the first stage was carried out in 38 districts, selected with probabilities proportional to their population size, then random samples were taken from the area, from the group of each area and household selection within the group, finally choosing the interviewee inside the house. People who had previously had a diagnosis of SARS-CoV-2 by RT-PCR or by serological test are excluded. The selection of a home was made, taking into account the next home at a distance of no less than 50 m and no more than five homes per block.

At the level of each of the selected households in each district, the following inclusion criteria were considered for the evaluation: person should be over 9 years of age and reside in the household for more than 21 days. Institutional places (residents of nursing homes, residential complexes, persons deprived of liberty, hospitalized persons, convents, and shelters) and health personnel who have worked in a healthcare center in the previous 30 days were excluded.

For the selection of the person, the Kish method was used, which consists of ordering all members of the family from the youngest to oldest, and then making a random selection through an electronic generator without repetition.

## SARS-Cov-2 antibody detection

The positivity of the antibodies against SARS-CoV-2 was determined by the lateral flow test (LFIA, Lateral Flow Immunoassay). "Coretest COVID-19 IgM/IgG Ab Test" from Core Technology, which uses the principle based on nano-colloidal technology, the application of a highly specific antigen-antibody reaction, and the principle of immunochromatographic analysis. The seropositivity in the present study is based on the qualitative detection of positive IgM, IgG, or IgM/IgG antibodies in all cases.

The tests were carried out using the brand protocol; the fingertip was pierced with a disposable lancet, collecting the blood with a pipette. Ten uL of whole blood plus two drops of detection buffer were added to the cassette, the results being read at 15 min.

## Procedures

The patient selection date was from June 24 2020 to July 10 2020. The fieldwork was carried out by trained health personnel, all under a strict selection and sampling protocol, as well as the correct application and interpretation of the test.

The selected participants were invited to participate in the study, after they accepted the epidemiological file elaborated by the Ministry of Health of Peru was applied, this file collects epidemiological data (age, sex, isolation, contact with a suspicious person or confirmed for SARS-CoV-2), clinical data (fever, cough, chills, sore throat, malaise, shortness of breath, headache, anosmia or ageusia, dysosmia, vomiting, diarrhea) as well as the presence of comorbidities (hypertension, diabetes, obesity, bronchial asthma, cancer, tuberculosis, COPD, etc.). The rapid test was taken at home, taking the appropriate biosecurity measures. The responses to the questionnaire and the test results were recorded on a website through a mobile application.

## Statistical analysis

Seroprevalence was estimated as the proportion of participants who had a positive rapid test result, whether IgM positive or IgG positive or IgM/IgG positive. An adjustment was made for the sampling weights used, having seroprevalence estimates for different selection probabilities. The sampling weights were calculated as the inverse of the probability of sampling selection within the size of each stratum.

Due to the complex design of the selection, it was taken into account that all statistical analyzes took into account the calculated weights, and this was carried out by analyzing with svyset in the statistical package Stata® v16.0 (StataCorp, College Station, TX, USA).

For the descriptive analysis, frequencies and percentages were estimated for categorical variables, and for numerical variables, the type of distribution was evaluated according to the histogram and evaluation of bias and kurtosis, being summarized in measures of central tendency and dispersion.

The bivariate analysis was carried out to know the relationship of factors to SARS-CoV-2 seropositivity, applying the chi-square test when the dependent variable was categorical, or with Student's *T* or Mann–Withney tests according to the distribution of the numeric variable.

Regression analysis was performed using generalized linear models, estimating prevalence ratios (PR) utilizing Poisson family and logarithm link, estimating 95% confidence intervals (95% CI) and *p*-values.

In addition, a correction was made for the prevalence of the disease due to the use of an inaccurate test (Sensitivity: 66% and specificity 97%) (*Lisboa Bastos et al., 2020*), using Bayesian estimation and the calculation of confidence intervals with Bootstrap.

## Ethical aspects

The research protocol was approved by the ethics committee of the Almanzor Aguinaga Asenjo Hospital (registration number 042-2020).

The people who were selected were explained the objective of the research, as well as the risks and benefits of the study. Written informed consent was obtained from all study participants. Consent for adults was used, as well as the consent and assent of children and adolescents, children under 9 years of age was not taken into the protocol so as not to violate their autonomy. Positive cases were notified to the coordinator for the application of outpatient management.

## RESULTS

Of the 2010 people surveyed, and estimated calculations by complex sampling, the seroprevalence at the regional level in Lambayeque of SARS-CoV-2 was 29.5% (95% CI [27.6–31.5]), and this was similar to the three provinces. See Table 1. The distribution of seroprevalences by district varied, reaching seroprevalences of 60% (See Fig. 1). After weighing the data by diagnostic performance, a prevalence of 42.15% was obtained (95% CI [38.17–46.51]).

The seroprevalence was similar in men and women (30.1% and 28.8% respectively). The age group from 31 to 40 years old presented the highest number of positive patients with 34.9%, this being lower in those older than 61 years and older, where it can be seen that seroprevalence decreased.

The presence of symptoms related to COVID-19 was reported by 22.8% of people, however, of them, only 43.5% gave a positive result. One in four asymptomatic people tested positive for SARS-CoV-2. Only 810 (41.4%) people complied with strict isolation in the region, and 25.8% of them were seropositive, as opposed to 32.2% of those who did not comply with strict isolation. See Table 1.

Of the patients with positive infection, symptoms, and signs the most frequent were dysgeusia (85.3%), dysosmia (83.6%), and diarrhea (60.5%). While those who had a negative result for SARS-CoV-2, the most frequent were irritability/confusion (30%), abdominal pain (29.3%), and nausea/vomiting (29.2%). See Fig. 2.

In the bivariate analysis, multiple symptoms and signs were associated with positivity for SARS-CoV-2, however, in the multivariate analysis, people who reported fever (PR: 1.41), general malaise (PR: 1.27), cough (PR: 1.44), dysosmia (PR: 1.69), chest pain (PR: 1.49) and back pain (PR: 1.45) were associated with a higher frequency of positivity. In the analysis of comorbidities, no comorbidity was associated with positivity for SARS-CoV-2.

Of the behaviors, those who reported complying with strict home isolation had 20% less frequency of testing positive for SARS-CoV-2 (PR = 0.80). However, having contact with patients with acute respiratory infection (ARI), contact with a confirmed case in the last 14 days, and going to the market, the frequency of testing positive was 60% (PR = 1.60), 51% (PR = 1.51 ) and 26% (PR = 1.26) higher; respectively. See Table 2.

## DISCUSSION

### Principal findings

Our study found that the seroprevalence of SARS-CoV-2 in the region of Lambayeque, Peru was 29.6%, where the province with the highest prevalence of the virus was

**Table 1 General characteristics and prevalence of SARS-CoV-2 in Lambayeque.**

| Variable | n (2010) | %* | Prevalence* (IgM/IgG +) | CI 95%* |
|---|---|---|---|---|
| Province | | | 29.5 | [27.6–31.5] |
| Lambayeque | 476 | 23.6 | 32.9 | [28.7–37.1] |
| Ferreñafe | 163 | 8.3 | 22.5 | [16.8–29.5] |
| Chiclayo | 1,371 | 68.1 | 29.2 | [26.9–31.6] |
| District | | | | |
| <20,000 hab. | 538 | 28.9 | 23.0% | [19.6–26.7] |
| 20,000–50,000 hab. | 469 | 25.6 | 32.7% | [28.7–37.0] |
| >50,000 hab. | 1,003 | 45.6 | 31.8% | [29.0–34.8] |
| Age group | | | | |
| 9–20 years | 220 | 11.1 | 26.9% | [21.4–33.2] |
| 21–30 years | 329 | 16.3 | 31.0% | [26.1–36.1] |
| 31–40 years | 394 | 19.5 | 34.9% | [30.3–39.8] |
| 41–50 years | 353 | 17.5 | 31.1% | [26.5–36.2] |
| 51–60 years | 315 | 15.7 | 29.8% | [25.0–35.1] |
| 61–70 years | 215 | 10.7 | 24.6% | [19.2–30.8] |
| 71–80 years | 129 | 6.4 | 20.3% | [14.2–28.2] |
| >80 years | 55 | 2.7 | 22.1% | [13.0–35.1] |
| Sex | | | | |
| Masculine | 949 | 52.7 | 30.1% | [27.4–33.0] |
| Femenine | 1,061 | 47.3 | 28.8% | [26.0–31.8] |
| Symptomatic | | | | |
| Yes | 440 | 22.8 | 43.6% | [38.9–48.3] |
| No | 1,570 | 77.2 | 25.4% | [23.3–27.5] |
| Isolation | | | | |
| Yes | 810 | 41.4 | 25.8% | [22.8–28.9] |
| No | 1,200 | 58.6 | 32.2% | [29.6–34.8] |
| Time of illness | | | | |
| ≤14 days | 239 | 55.2 | 33.3% | [27.5–39.6] |
| >14 days | 201 | 44.8 | 56.2% | [49.2–63.0] |
| A previous visit to healthcare facility | | | | |
| Yes | 67 | 3.4 | 39.1% | [28.1–51.3] |
| No | 1,943 | 96.6 | 29.2% | [27.2–31.2] |
| Contact with ARI case | | | | |
| Yes | 123 | 5.9 | 57.2% | [48.2–65.7] |
| No | 1,887 | 94.1 | 27.8% | [25.8–29.8] |
| Contact with a confirmed case | | | | |
| Yes | 142 | 7.2 | 52.3% | [44.0–60.5] |
| No | 1,868 | 92.8 | 27.7% | [25.8–29.8] |
| Visit a market | | | | |
| Yes | 442 | 21.1 | 38.0% | [33.6–42.6] |
| No | 1,568 | 78.9 | 27.1% | [25.1–29.5] |
| Co-morbidity | | | | |
| Yes | 450 | 22.3 | 27.4% | [23.4–31.7] |
| No | 1,560 | 77.7 | 30.3% | [27.9–32.4] |

**Note:**
* Estimates obtained by complex sampling. Strata were district.

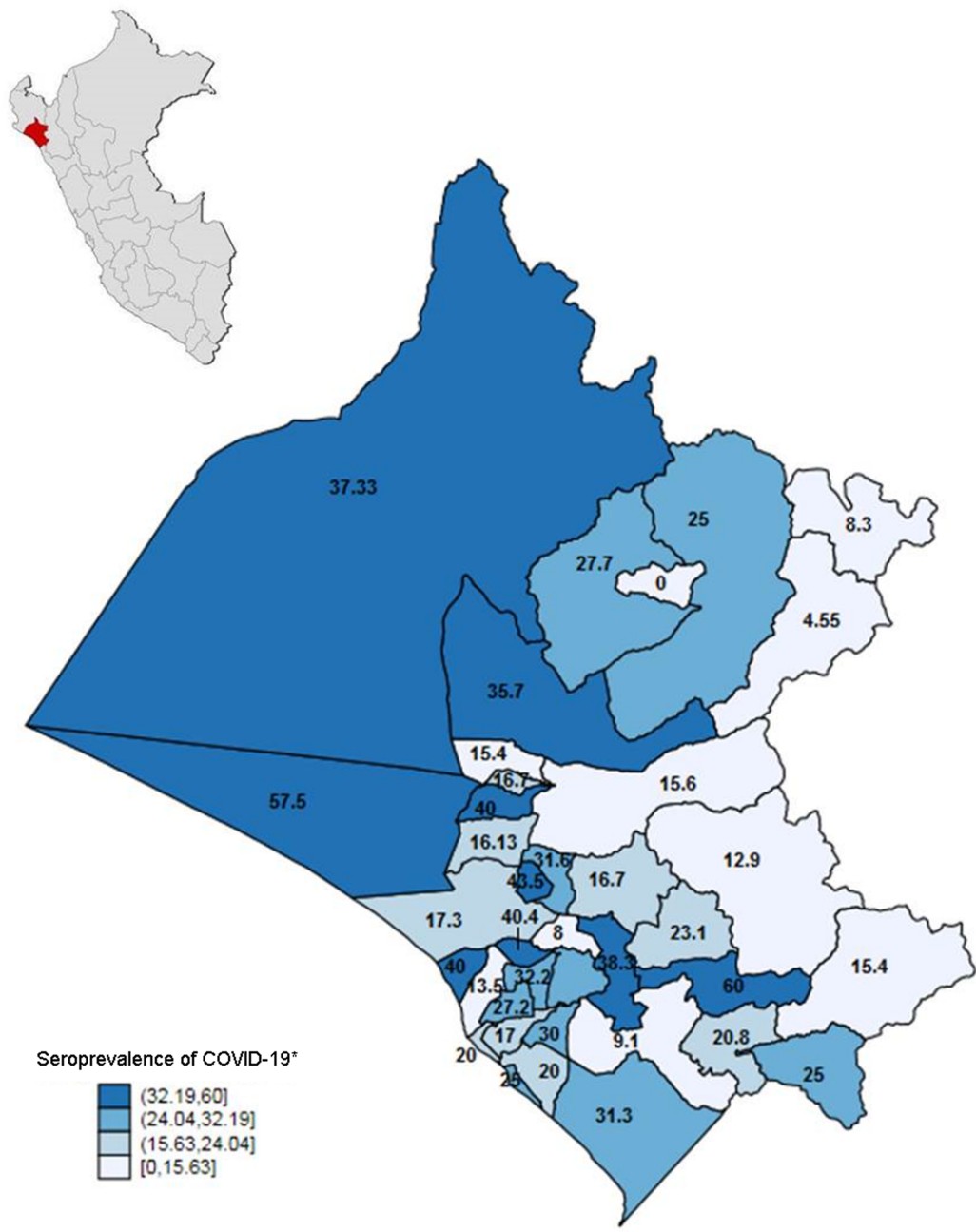

**Figure 1** **Geospatial distribution of seroprevalence of SARS-CoV-2 in Lambayeque, Peru.** *Estimates obtained by complex sampling. Strata were district.

Lambayeque (31.7%), followed by Chiclayo (29.8%). 28.1% presented an active infection. Almost one in five patients was asymptomatic (25.7%) and the most frequently reported symptoms were cough (29.7%), headache (29.4%), and sore throat (26.1%).

Additionally, we also explored the epidemiological clinical factors related to the seroprevalence of SARS-CoV-2. The epidemiological factors associated with a higher seroprevalence of SARS-CoV-2 were reported having had contact with a person with ARI,

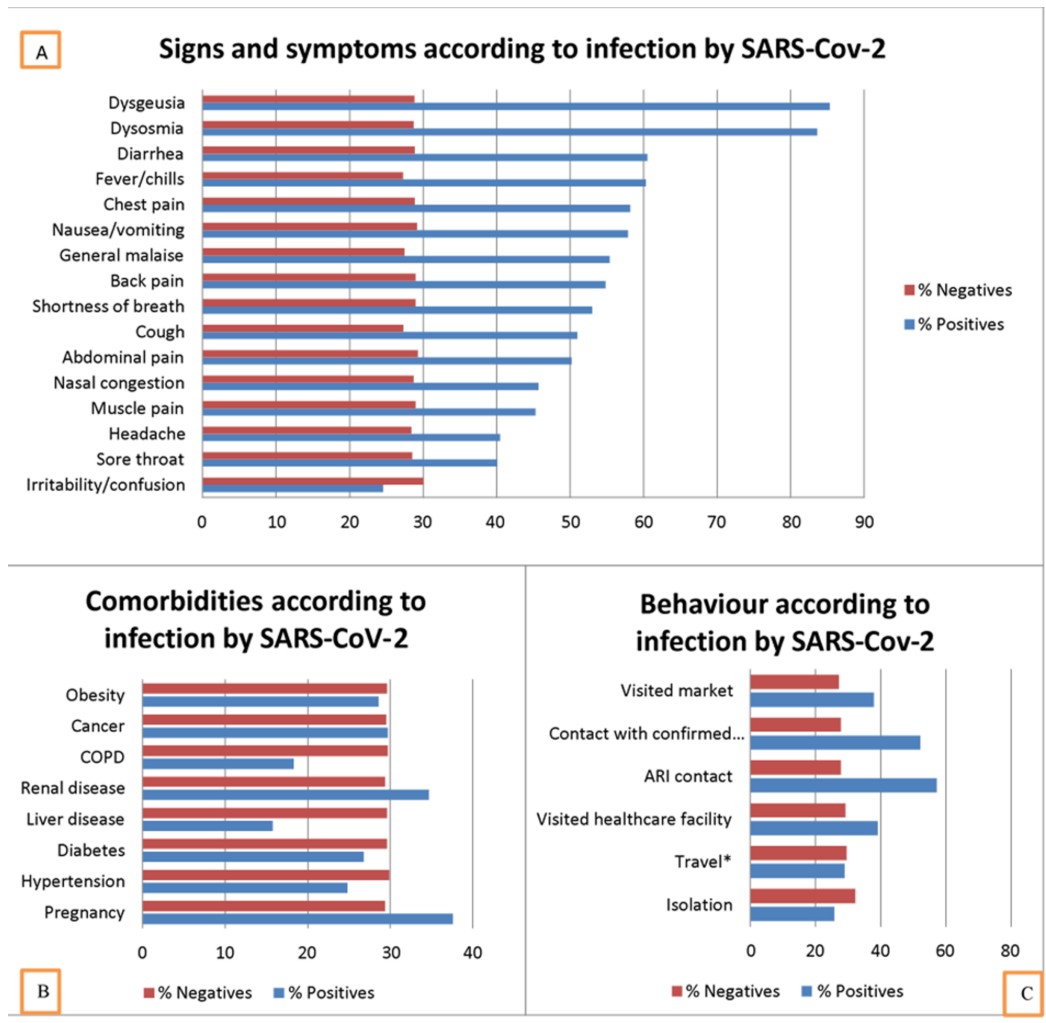

**Figure 2 Characteristics according to SARS-CoV-2 positivity in Lambayeque, Peru.** COPD: Chronic obstructive pulmonary disease. ARI: Acute respiratory infection. *Estimates obtained by complex sampling. Strata were district. (A) Signs and symptoms according to infection by SARS-CoV-2. (B) Comorbilities according to infection by SARS-CoV-2. (C) Behaviour according to infection by SARS-CoV-2. Blue percent with positives. Red percent without SARS-CoV-2.

contact with a confirmed case, and having visited a market 14 days before taking a sample to detect the virus. Regarding clinical factors, it was found that people who reported fever, general malaise, cough, dysosmia, abdominal pain, chest pain, and back pain had a higher seroprevalence of SARS-CoV-2.

## Seroprevalence of SARS-CoV-2

The SARS-CoV-2 seroprevalence study in the Lambayeque region, Peru showed that 29.6% (95% CI [27.5–31.6]) of the population present anti-SARS-CoV-2 antibodies, and it is estimated that 354,389 infected people. This is higher than the official report of the epidemiological situation reported during the investigation period (1,197,260 habitants;

**Table 2 Characteristics associated with seropositivity for SARS-CoV-2 in Lambayeque, Peru.**

| Characteristics | PRc | CI 95% | PR(a) | CI 95% with $p$-values |
|---|---|---|---|---|
| Signs/symptoms | | | Model I | |
| Fever/chills | 2.21 | [1.89–2.59] | 1.41 | [1.09–1.82] $p = 0.008$ |
| General malaise | 2.02 | [1.71–2.38] | 1.28 | [0.99–1.65] $p = 0.059$ |
| Cough | 1.87 | [1.59–2.20] | 1.44 | [1.16–1.78] $p = 0.001$ |
| Sore throat | 1.40 | [1.15–1.72] | 0.92 | [0.73–1.18] $p = 0.511$ |
| Nasal congestion | 1.59 | [1.26–2.09] | 1.05 | [0.80–1.38] $p = 0.728$ |
| Shortness of breath | 1.83 | [1.36–2.45] | 0.83 | [0.57–1.21] $p = 0.340$ |
| Diarrhea | 2.10 | [1.62–2.71] | 1.12 | [0.83–1.53] $p = 0.439$ |
| Nausea/vomiting | 1.99 | [1.39–2.85] | 1.16 | [0.74–1.80] $p = 0.537$ |
| Headache | 1.43 | [1.17–1.74] | 0.89 | [0.71–1.12] $p = 0.339$ |
| Irritability/confusion | 0.83 | [0.35–1.98] | 0.40 | [0.20–0.79] $p = 0.009$ |
| Dysosmia | 2.92 | [2.45–3.46] | 1.69 | [1.15–2.48] $p = 0.007$ |
| Dysgeusia | 2.96 | [2.49–3.52] | 1.22 | [0.84–1.78] $p = 0.304$ |
| Muscle pain | 1.56 | [1.18–2.08] | 1.02 | [0.76–1.39] $p = 0.880$ |
| Abdominal pain | 1.71 | [1.07–2.74] | 1.14 | [0.70–1.83] $p = 0.602$ |
| Chest pain | 2.02 | [1.54–2.64] | 1.49 | [1.09–2.05] $p = 0.014$ |
| Backache | 1.89 | [1.41–2.53] | 1.45 | [1.05–1.99] $p = 0.022$ |
| Co-morbidity | | | Model II | |
| Pregnancy | 1.28 | [0.73–2.22] | 1.25 | [0.72–2.17] $p = 0.429$ |
| Hypertension | 0.83 | [0.62–1.11] | 0.84 | [0.62–1.13] $p = 0.246$ |
| Diabetes | 0.90 | [0.65–1.26] | 0.93 | [0.66–1.30] $p = 0.663$ |
| Liver disease | 0.53 | [0.15–1.89] | 0.55 | [0.16–1.92] $p = 0.344$ |
| Renal disease | 1.18 | [0.69–2.02] | 1.19 | [0.69–2.03] $p = 0.533$ |
| COPD | 0.62 | [0.28–1.37] | 0.63 | [0.28–1.39] $p = 0.930$ |
| Cancer | 1.01 | [0.44–2.30] | 1.04 | [0.45–2.40] $p = 0.250$ |
| Obesity | 0.97 | [0.75–1.24] | 0.96 | [0.75–1.24] $p = 0.770$ |
| Behavior | | | Model III | |
| Isolation | 0.80 | [0.69–0.93] | 0.80 | [0.69–0.92] $p = 0.273$ |
| Travel* | 0.98 | [0.68–1.42] | 0.79 | [0.54–1.16] $p = 0.472$ |
| Been in healthcare facility* | 1.34 | [0.98–1.83] | 1.12 | [0.80–1.55] $p = 0.000$ |
| Contact with ARI* | 2.06 | [1.74–2.44] | 1.60 | [1.27–2.00] $p = 0.001$ |
| Contact with confirmed case (14 previous days) | 1.88 | [1.58–2.25] | 1.51 | [1.21–1.89] $p = 0.001$ |
| Been to market* | 1.40 | [1.21–1.61] | 1.26 | [1.09–1.46] $p = 0.001$ |

Notes:
* In the 14 days before taking the sample.
ARI: Acute respiratory infection. COPD: Chronic obstructive pulmonary disease.
Estimation carried out by complex samplings, prevalence ratios obtained with generalized linear models with Poisson family, and logarithmic link. PR: Prevalence ratio. C: raw, a: adjusted. CI: Confidence interval.
Model I: Number of strata = 38; Number of PSUs = 2,010; Degree of freedom (df) = 1,972; $F(16,1957) = 22.59$; Prob > $F = 0.000$.
Model II: Number of strata = 38; Number of PSUs = 2,010; Degree of freedom (df) = 1,972; $F(8,1965) = 0.65$; Prob > $F = 0.7316$.
Model III: Number of strata = 38; Number of PSUs = 2,010; Degree of freedom (df) = 1,972; $F(5,1968) = 20.02$; Prob > $F = 0.0000$.

1.1% positivity; 14,355 infected people; 5,986 dead, source: Lambayeque epidemiological room July 10, 2020). The infection fatality ratio calculated was 0.5%.

This finding, to our knowledge, suggests that the Lambayeque region represents the highest seroprevalence of SARS-CoV-2. To date, the closest to this seroprevalence is in Delhi, India (23.48%) (*Ministry of Health and Family Welfare of India, 2020*) where they evaluated from June 27 to July 10, 2020, and only considered the prevalence of positive IgG. Other studies indicate lower seroprevalences such as Madrid-Spain (11.3%, 95% CI [9.8–13]) (*Pollán et al., 2020*), Geneva-Switzerland (4.8%, 95% CI [2.4–8.0] to 10.8%, 95% CI [8.2–13.9]) (*Stringhini et al., 2020*), France (6.7%, 95% CI [5.36–8.11]), Los Angeles-California (4.65%, 95% CI [2.52–7.07]) (*Sood et al., 2020*), China (3.2–3.8%) (*Xu et al., 2020*), Italy (2.5%, 95% CI [2.3–2.6]), Georgia, USA (2.5%; 95% CI [1.4–4.45]) (*Biggs et al., 2020*), Indiana, USA (1.1%, 95% CI [0.76–2.54]) (*Menachemi et al., 2020*).

Technical reports of seroprevalence studies led by Regional Health Directorates (DIRESA), still without being published in scientific publications, affirm figures lower than those found in Lambayeque: Lima, Peruvian capital (24.3%, 95% CI [18.5–24.3]), Cusco, southern Peruvian region (2.65%, 95% CI [1.23–4.07]). However, in Iquitos, a seroprevalence of 71% was found (95% CI [68.0–74.7]). The differences found between the estimates of the regions evaluated could be due to the date of the research (peaks between regions), differences in the sample size and type of sampling used, among others. It should be mentioned that all of these Peruvian seroprevalence studies for the detection of SARS-CoV-2 used the rapid immunochromatography test.

The high seroprevalence reported in our research could be due to the identification of epidemiological factors in the population of our study that condition high transmissibility of the virus in the region. This is because 58.6% reported that they have not maintained the mandatory social isolation measures (isolation PR: 0.80) and 21.1% have visited the market 14 days prior to the investigation (PR: 1.40), and this conditions high transmissibility of the virus in the region. An investigation in Peru found that the frequency of COVID-19 in market workers ranged from 20% to 90%, so these places were areas of contagion in Lambayeque (*Iglesias-Osores, Saavedra-Camacho & Córdova-Rojas, 2020*). Another factor that could explain the differences found is the method used to detect antibodies against SARS-CoV-2, which has lower sensitivity in contrast to ELISA or chemiluminescence tests (*Lisboa Bastos et al., 2020*). Our study used the immunochromatography method, in contrast to that reported in India, the United States, and Switzerland, where antibodies were measured with ELISA and in Italy (ELISA and/or chemiluminescence-CLIA).

Regarding the seroprevalence reported in the districts of Lambayeque, we found a geographic variability in the estimates. The districts of Picsi, Zaña, Kañaris, Incahuasi, Chochope presented a seroprevalence of less than 10%. On the other hand, seroprevalences higher than 50% were estimated in the districts of Pucalá and Mórrope. The geographic variability found in our study is consistent with that reported in the United States (*Havers et al., 2020*), where the seroprevalence range fluctuated between 1% and 6.9%.

This could be due to the great geographic variability that exists in Lambayeque since each district has different overcrowding, behavior before the disease and level of

development; it may also be because some conglomerates have probably respected the social isolation measures decreed by the government during the state of emergency, which would result in lower transmissibility of the virus in those districts. By Peruvian regulations, all inhabitants when leaving their house had to wear a mask and have physical distance from other people. It is necessary to mention the geographical point in which the Lambayeque region is located, which is an intermediate point in Peruvian territory with close access to multiple areas of the coast, mountains, and jungle; which results in a very active formal and informal economic activity, as well as an easy point of entry for probable asymptomatic or symptomatic residents of COVID-19. Finally, the high prevalence could be because this region was severely hit by the high transmissibility of the virus that caused the near sanitary collapse at the highest peak of the SARS-CoV-2 infection.

The estimated prevalence against SARS-CoV-2 was similar between both sexes (men: 29.2%; women: 30.0%). This is consistent with what is described in the technical reports made in our country (Lima and Cusco) and the population findings in Spain, the United States (*Havers et al., 2020*; *Pollán et al., 2020*). However, it differs from what was reported in Iquitos, where 73% and 58% of women and men had antibodies against SARS-CoV-2s, respectively, this difference could be due to the fact that, in Iquitos, the proportion of women and men is 4:3.

## Epidemiological factors associated with the seroprevalence of SARS-CoV-2

We found that people who reported respecting home isolation had a lower SARS-CoV-2 seroprevalence compared to those who mentioned having left their home during the health emergency period. This is consistent with what has been evidenced in studies conducted in Europe (*Kuitunen et al., 2020*; *Martínez-Valero, Miranda & Martín-Sánchez, 2020*). This could be explained by the fact that the effect of isolation and physical distance slows down the transmission of the disease, especially this viral disease that is easily transmissible. A recent review found that physical isolation measures and the use of masks are effective in reducing the transmission of the disease (*Chu et al., 2020*).

While reporting has been in contact with a person with ARI and a confirmed COVID-19 case 2 weeks before participation in this research was associated with a higher prevalence of SARS-CoV-2 seroprevalence. This has been corroborated in a study that evaluated close contacts who lived together with patients infected with SARS-CoV-2, who had up to six times the risk of contracting the disease compared to other not so close cases (*Bi et al., 2020*).

Likewise, going to a market 2 weeks before taking a sample for detection of SARS-CoV-2 represented a higher seroprevalence of SARS-CoV-2. In our search for information, we only found one study conducted in Baodi, China where 75.5% of its confirmed cases had gone to markets (*Wu et al., 2020*). The probable explanation is that the markets that exist in Lambayeque have not had a strict control to monitor compliance with biosafety protocols, reduce their capacity, and it was unknown whether

workers and/or buyers were infected, thus that would increase the possibility of transmission as has been explained.

## Clinical factors associated with SARS-CoV-2 seroprevalence

Participants who reported chest pain, back pain, fever, malaise, and cough were associated with higher SARS-CoV-2 seroprevalence. This is similar to that reported in other studies on factors related to virus positivity (*Hu et al., 2020*; *Li et al., 2020*), it should be noted that, unlike other studies where fever and cough are the most frequent symptoms (88.5% and 68.5% respectively (*Li et al., 2020*)), our study found that the most frequent symptoms were cough and headache (29.7% and 29.4% respectively). This very marked difference may be because the studies where information on signs and symptoms is available are from hospital studies, unlike this study that was carried out at the community level.

## Implications of the findings

This research provides consistent scientific evidence for improving health decision-making aimed at mitigating the impact of the COVID-19 pandemic in the Lambayeque region. These findings represent the baseline measurement that will subsequently make it possible to know the proportion of participants negative for SARS-CoV-2 who acquire active disease infection, after completing the clinical follow-up. Additionally, it allows addressing the frequency of asymptomatic cases and subsequently identifying the presentation of symptoms throughout the clinical follow-up. These findings will also make it possible to focus efforts on the districts with the highest proportion of seroprevalence (epidemiological fences, traceability of cases and contacts at the community level, targeted mandatory social quarantine measures, reinforcement, and expansion of the first level of care and rapid response teams, among others).

## Limitations and strengths

Our research has some limitations. First, it was not possible to use the immunoassay or immunochemiluminesence diagnostic method to determine IgG antibodies, given its better diagnostic precision, so the real estimate may have been much higher, as reported in the correction with a Bayesian sample. Second, the seroprevalence found represents only one Peruvian region, a country characterized by having multiple epidemics due to its wide geographic diversity; however, these results may be representative since an estimation has been made with complex multistage sampling and stratified by sex. Third, there is a certain information bias in the estimates of factors associated with seroprevalence, since it has not been possible to explore other characteristics that influence the high transmissibility of the virus (economic level, educational level, overcrowding, and basic services in housing, among others) with which may be an underestimation of the measured factors, and there may be other unmeasured factors. Fourth, the actual seroprevalence may be underestimated as repeated antibody measurements could not be performed at different time points. Fifth, a double serological test was not performed to differentiate a true and false positive; however, an attempt has been made to control the

error in the test, so a Bayesian analysis was added to the results. Sixth, by not including children less than 9 years of age, the study may not be representative of the entire population. However, to our knowledge, this research represents one of the first population seroepidemiological studies with a longitudinal design, even at the Latin American level, characterized by a rigorous methodological implementation to conduct the research, particularly in aspects of sampling, quality of information collection by government-trained rapid response teams, sampling and biostatistical processing.

## CONCLUSIONS

This research suggests that almost one in three people residing in the Lambayeque region, Peru have antibodies against SARS-CoV-2. While having contact with a person with ARI, contact with a confirmed case, going to a market the previous 14 days, presenting general discomfort, cough, dysosmia, abdominal pain, chest pain, back pain could be related to a greater seroprevalence of SARS-CoV-2.

## ACKNOWLEDGEMENTS

We thank the rapid response teams of EsSalud and "Gerencia Regional de Salud de Lambayeque", for the support in the execution and "Comando de Operaciones Regional Lambayeque Sipan" for the technical support in the execution. We thank EsSalud and "Gerencia Regional de Salud de Lambayeque" for support with the serological tests used.

### Funding

This work was supported by EsSalud and "Gerencia Regional de Salud de Lambayeque". The funders had no role in study design, data collection and analysis, decision to publish, or preparation of the manuscript.

### Grant Disclosures

The following grant information was disclosed by the authors:
EsSalud and "Gerencia Regional de Salud de Lambayeque".

### Competing Interests

The authors declare that they have no competing interests.

### Author Contributions

- Cristian Díaz-Vélez conceived and designed the experiments, performed the experiments, authored or reviewed drafts of the paper, and approved the final draft.
- Virgilio E. Failoc-Rojas conceived and designed the experiments, analyzed the data, prepared figures and/or tables, authored or reviewed drafts of the paper, and approved the final draft.
- Mario J. Valladares-Garrido conceived and designed the experiments, analyzed the data, authored or reviewed drafts of the paper, and approved the final draft.

- Juan Colchado conceived and designed the experiments, performed the experiments, authored or reviewed drafts of the paper, and approved the final draft.
- Lourdes Carrera-Acosta conceived and designed the experiments, authored or reviewed drafts of the paper, and approved the final draft.
- Mileny Becerra conceived and designed the experiments, authored or reviewed drafts of the paper, and approved the final draft.
- Dafne Moreno Paico conceived and designed the experiments, performed the experiments, authored or reviewed drafts of the paper, and approved the final draft.
- Elgin Thom Ocampo-Salazar conceived and designed the experiments, performed the experiments, authored or reviewed drafts of the paper, and approved the final draft.

### Ethics

The following information was supplied relating to ethical approvals (i.e., approving body and any reference numbers):

The research protocol was approved by the ethics committee of the Almanzor Aguinaga Asenjo Hospital (registration number 042-2020).

### Data Availability

The study's database and codebook are available in the Supplemental Files.

### Supplemental Information

Supplemental information for this article can be found online at http://dx.doi.org/10.7717/peerj.11210#supplemental-information.

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
