# Peer review of "SARS-CoV-2 seroprevalence study in Lambayeque, Peru. June–July 2020"

_PeerJ, doi:10.7717/peerj.11210_

## Round 0.1 · original submission · Major Revisions

Dear Dr. Failoc-Rojas,
The reviewers have provided good comments on your work but raised a question of validation by RT-PCR or some other technique. Reviewer 2 has also raised some points on rigor. I think these issues should be addressed and limitations should be mentioned upfront. I invite you to submit the revised version of the manuscript answering the reviewer's comments.

Sincerely,

Gunjan

Reviewer 1 ·

Basic reporting

The Manuscript is clear and easy to understand.

Experimental design

Experimental design was good but I feel there was scope of improvement.

Validity of the findings

Conclusions are well stated and according to the study's finding.

Additional comments

Dear Authors,
First of all, I would like to congratulate you for the fantastic work you have done in this study. The manuscript by Diaz-Velez et al. discussed the serological surveillance, based on the measurement of the presence specific antibodies (the test kit used in this study was approved by FDA) against SARS-CoV-2 in a given population in Lambayeque region of Peru. Population-based serosurveys are a powerful source of information to monitor the progression of the epidemic. The authors have examined the presence of IgG/IgM antibodies from 2010 people. The results and conclusions of the papers are interesting and important. I also find the study to be relevant and well written. I liked reading this paper, and the text description of the results was convincing. However, I found some concerns that need to be addressed before considering this manuscript for publication.
1. Why were the individuals below the age of 9 not included in the study? This study may not be the representative of the population.
2. The percentage of participants that had previous of SARS-CoV-2 PCR testing was not reported in this study therefore it is not clear whether or not there was a bias for probable cases.
3. What measures had been taken to differentiate between a true and false positive?
4. I am wondering, were the authors considered repeated measurements of antibodies at different time points for the seroprevalence estimation? Antibody titers may decline with time and this would give falsely low prevalence estimates.
5. It could have been better if the authors would consider the gender, body mass index and smoking habit as a trait for figure 2C.
6. Did the authors estimate the infection-to-fatality ratio of COVID-19 in the given population?

Reviewer 2 ·

Basic reporting

- Please cite a reference for lines 48-50.
- Lines 60-64 need to be divided into multiple sentences. The sentence construction is not ideal.
- The introduction section needs more information. This includes: What is the prevalence of false positives in the serological COVID tests? What is the population division by age, gender, preexisting conditions in the Lambayeque region? What are the possible causes for almost 88.7% of Peru’s cases being asymptomatic?
- Line 241 should be “this finding” and not “his finding”. Please correct.

Experimental design

This was an exhaustive study in a set-up which is ever changing and therefore difficult to conclude. The authors have done a commendable job considering the limitations, however there are some major aspects which have not been listed in the experimental design:

- For the selected population, what percentage had preexisting conditions asper the collected questionnaire? Was this confirmed with medical records of the patients?
- Both IgG and IgM positive results were considered. Explain how the results between the 2 antibodies could vary and how they could lead to different conclusions.
- For individuals who did not practice strict isolation, was the use of masks taken into consideration? How did that affect the results?
- Were any of the seropositive individuals tested with qPCR too? Would that have helped in having greater faith in the results?
- Was an orthogonal testing algorithm as recommended by the CDC employed for this study?

Validity of the findings

- The findings of the study though interesting, are based on experimental design which has scientific omissions and lack of rigor. Hence it is difficult to look at the findings without taking these errors into consideration.
- Statistical analysis is thorough and reliable.

Additional comments

Listed in the sections above.

Reviewer 3 ·

Basic reporting

The article describes the rationale and the methodology of the study in sufficient detail. The message that the article wants to convey can be much easier to elucidate if grammatical errors are addressed. For instance, lines 59-64, 134-142 needs to be rephrased. Authors should also address the typographical errors in the text.

Experimental design

The experimental design is well-thought-of. It will be nice if the authors could show some kind of validation for some of the positive tested samples using conventional qPCR. This will help better in assessing the false-positive or false-negative rates.

Validity of the findings

The authors have not commented on the false positive and false negative rates for their analysis. It has been reported that some of these tests do have a kind of false positivity or negativity. Did the authors observe this in their analysis?

Additional comments

Authors have very nicely demonstrated the SARS-CoV-2 seroprevalence study in Lambayeque, Peru in the manuscript and have reported their analysis following proper statistical tests. The manuscript is suitable for publication, provided the above comments are addressed.

---

## Round 0.2 · Minor Revisions

Please answer Reviewer 4's comments.

Reviewer 1 ·

Basic reporting

The Manuscript has improved considerable after the authors considered the reviewer's suggestion.

Experimental design

Authors were acknowledged the limitations of the study in the revised version.

Validity of the findings

Conclusions are well stated and according to the study's finding

Reviewer 2 ·

Basic reporting

The language has been corrected where necessary and typographical errors have been addressed.
Background section is robust and authors have duly added information that was previously asked.

Experimental design

Robust experimental design. The lack of qPCR testing has been considered as a limitation.

Validity of the findings

Data is statistically sound and adequate controls have been used.

Additional comments

The authors have a done a sound job with addressing previous concerns.

Reviewer 4 ·

Basic reporting

The language can be better

Experimental design

It needs some more rigorous analysis as I will highlight later

Validity of the findings

This is a well done paper but could be made better.

Additional comments

The authors do a good job of rebutting the reviewers. However, I would like to ask some more analysis from the authors
1) a)"The high seroprevalence reported in our research could be due to the identification social isolation measures and 27.1% have visited a market 14 days prior to the investigation. "
b) "This is because 25.9% reported that they have not maintained the mandatory that could explain the differences found is the method used to detect antibodies against SARS-CoV-2, which has lower sensitivity in contrast to ELISA or chemiluminescence tests (Lisboa Bastos et al. 2020)."
Can the authors support both the above statements with more rigorous analysis as opposed to just percentages? This analysis can then extended to other such factors which can explain the high seroprevalance.
2) How much did the past medical history of the sample size influence SARS-COV2 positive test rate? Is it possible that the general population was less healthier before SARS-COV2, which caused more people to fall sick, and thus the high seroprevalance?

Minor comment
1) Language should be brushed up

---

## Round 0.3 · accepted · Accept

The authors have answered all reviewer comments satisfactorily.

Reviewer 4 ·

Basic reporting

No issues

Experimental design

Good

Validity of the findings

Good

Additional comments

The authors did not quite respond to the issue I raised especially point 1. However, the manuscript is good and can be accepted.